# Fitness consequences of depressive symptoms vary between generations: Evidence from a large cohort of women across the 20th century

Christopher I. Gurguis[1]*, Renée A. Duckworth[2], Nicole M. Bucaro[1], Consuelo Walss-Bass[1]

1 Department of Psychiatry and Behavioral Sciences, McGovern Medical School at UTHealth, Houston, TX, United States of America, 2 Department of Ecology and Evolutionary Biology, University of Arizona, Tucson, AZ, United States of America

* christopher.gurguis@uth.tmc.edu

## Abstract

Depression has strong negative impacts on how individuals function, leading to the assumption that there is strong negative selection on this trait that should deplete genetic variation and decrease its prevalence in human populations. Yet, depressive symptoms remain common. While there has been a large body of work trying to resolve this paradox by mapping genetic variation of this complex trait, there have been few direct empirical tests of the core assumption that there is consistent negative selection on depression in human populations. Here, we use a unique long-term dataset from the National Health and Nutrition Examination Survey that spans four generational cohorts (Silent Generation: 1928–1945, Baby Boomers: 1946–1964, Generation X: 1965–1980, and Millenials: 1981–1996) to measure both depression scores and fitness components (lifetime sexual partners, pregnancies, and live births) of women from the United States born between 1938–1994. We not only assess fitness consequences of depression across multiple generations to determine whether the strength and direction of selection on depression has changed over time, but we also pair these fitness measurements with mixed models to assess how several important covariates, including age, body mass, education, race/ethnicity, and income might influence this relationship. We found that, overall, selection on depression was positive and the strength of selection changed over time–women reporting higher depression had relatively more sexual partners, pregnancies, and births except during the Silent Generation when selection coefficients neared zero. We also found that depression scores and fitness components differed among generations—Baby Boomers showed the highest severity of depression and the most sexual partners. These results were not changed by the inclusion of covariates in our models. A limitation of this study is that for the Millenials, reproduction has not completed and data for this generation is interrupted by right censoring. Most importantly, our results undermine the common belief that there is consistent negative selection on depression and demonstrate that the relationship between depression and fitness changes between generations, which may explain its maintenance in human populations.

**Data Availability Statement:** All data are available for download from the NHANES databases and there is no unique accession number. The data was accessed from the NHANES site for the years

mentioned in our manuscript from the following link: https://wwwn.cdc.gov/nchs/nhanes/Default.aspx.

**Funding:** The author(s) received no specific funding for this work.

**Competing interests:** The authors have declared that no competing interests exist.

## Introduction

Understanding the maintenance of genetic variation in traits under consistent selection is a classic problem in evolutionary biology [1]. Mental illness is commonly claimed to represent an example of this problem because many disorders are common, heritable, and harmful. Under consistent negative selection, genetic variation for mental illness and its prevalence in populations should decrease [2]. Yet, over the 20th century, the prevalence and heritability of mental illness has increased [3,4]. This pattern is inconsistent with the generally held assumption that natural selection against mental illness is strong. Further complicating resolution of this paradox are different concepts of harm in evolutionary biology and psychiatry. In evolutionary biology, harm refers to negative fitness consequences, which are rarely measured for mental illness. In psychiatry, harm refers to functional impacts in one's professional or personal domains but does not necessarily entail fitness loss.

Evolutionary psychiatrists have proposed several possible mechanisms to resolve this apparent paradox, including neutral evolution, fluctuating selection, and mutation-selection balance [2]. These solutions are largely based on verbal arguments and narrative reviews of the literature. While no explicit empirical tests of these mechanisms have been published, a few studies have examined the fitness consequences of various psychiatric disorders. In most studies, the common approach is to compare fitness components of individuals affected by mental illness to those of their unaffected siblings or the general population. Those diagnosed with bipolar disorder and schizophrenia show decreased fecundity [5–7], whereas the relationship between fecundity and diagnoses of depressive or anxiety disorders is more complex [8–12]. Only one study examined the quantitative relationships between psychiatric symptoms and fitness components and they found that anxiety had a positive relationship with the number of children, grandchildren, and great-grandchildren 15 years later [10]. This lack of evidence is problematic since depressive symptoms measured in the general population are known to be heritable–a key component in predicting response of a trait to selection [13].

Evolutionary studies of mental illness that focus on explaining the persistence of clinical syndromes (e.g. schizophrenia, bipolar disorder, major depressive disorder) examine extreme expressions of the traits involved and ignore the fact that many symptoms (e.g. low mood) vary continuously within populations. Coding mental illness as a dichotomous phenotype is problematic because selection on a trait is the covariance between that trait and fitness and focusing on individuals only at the extremes of a trait distribution (i.e. those with diagnosed mental illness) can result in misleading or spurious relationships [14,15]. Understanding the fitness consequences of mental illness depends on accurately measuring the relationship between psychiatric symptoms and fitness across the population, rather than focusing on clinically diagnosed syndromes which are arbitrarily defined with respect to biology and fitness and may miss being diagnosed depending on individual access to health care [16].

Depression provides a unique opportunity to study the evolution of mental illness because the symptoms of depression are expressed widely across the population. Depressive disorders affect 4.4% of the global population and are a leading cause of disability worldwide [17]. The lifetime prevalence of major depressive episodes may be as high as 20% and these episodes may occur as part of different psychiatric disorders [18]. The symptoms of major depressive episodes are also expressed in individuals unaffected by mental illness and are readily measured with self-report screening, allowing for a quantitative measurement of depressive symptoms across the population [19]. Here, we use a large dataset from the National Health and Nutrition Examination Survey (NHANES) spanning four generational cohorts of women from across the United States (U.S.) to test the widely held assumption that mental illness has negative fitness consequences and to assess the possibility that variation in depressive

symptoms is maintained through fluctuations in natural selection. Of note, while this dataset spans over 50 years and 4 generations, the last generation included (see definitions below) is still reproducing and therefore data collected from them is interrupted by right censored. We discuss this limitation and its implications for interpretation of results in detail below (see Discussion).

## Methods

### National health and nutrition examination survey

The NHANES is a nationally representative survey of the U. S. population that began in 1959. Data is collected from approximately 5,000 participants yearly and includes an interview, physical examination, and collection of laboratory specimens. The interview portion includes demographic data and questions regarding sexual behavior, reproductive health, and mental health of participants as well as the number of pregnancies and live births reported by women.

Since 2005, the Patient Health Questionnaire-9 (PHQ-9), a reliable and valid screen for depression, had been added to this survey and was collected in participants at a single time point [19]. Although repeated measures of PHQ-9 are not collected in NHANES, a recent meta-analysis demonstrated high repeatability of the measure over periods of up to two weeks with limited data over longer periods [20]. Examination of NHANES data has also demonstrated that PHQ-9 score distribution is relatively stable with age and shows invariance for sex, race/ethnicity, and education level, allowing for meaningful comparisons of the measure through time and across major U.S. sociodemographic groups [21,22]. Single nucleotide polymorphism (SNP) heritability ranges from 6% to 9% for various components of the PHQ-9, which are moderately to strongly genetically correlated with each other [23].

Data on all study subjects was downloaded from the NHANES database, which is publicly available. Detailed information on methods for recruitment, measurement and evaluation techniques, and the informed consent process can be found on the NHANES website (https://www.cdc.gov/nchs/nhanes/index.htm). All data analyzed by the authors of the current study was deidentified and authors had no way of identifying study subjects based on the data.

### Inclusion and exclusion criteria

For this study, we obtained data collected as part of NHANES between 2005–2016. Participants included women age 20–69 (birth years spanning 1938–1994). In NHANES datasets, gender is self-reported and gender identity is not distinguished from sex assigned at birth. PHQ-9 was administered to participants >18-years-old. We did not include participants older than 69 because information on number of male sexual partners was not collected for respondents >69-years-old. Similarly, because some information important for analyses (e.g. education level) was not collected on respondents <20-years-old, we used age 20 as the lower age limit. Individuals with a history of intellectual or developmental disability or other developmental problems were excluded from analyses, but not individuals with prior history of psychiatric disorder(s). In total, 12,468 women completed the PHQ-9. Of these respondents, 10,030 women reported their lifetime number of male sexual partners, pregnancies, and live births as well as important covariates (BMI, race/ethnicity, education history, and family income). Importantly, relatedness of these individuals is not assessed and cannot be included in our analyses. A study of NHANES III (1988–1994) and NHANES data from 1999–2002 did reveal that cryptic relatedness may be a limitation of this dataset, though the extent to which this is problematic for the present study (which examines data collected over a span of different years) is not known [24].

## Definition of generational cohorts

Respondents were divided into four generational cohorts based on their birth year: the Silent Generation (individuals born between 1928–1945), the Baby Boomer Generation (individuals born between 1946–1964), Generation X (individuals born between 1965–1980), and the Millennial Generation (individuals born between 1981–1996). Because there is high generational overlap in humans, the definition of generations is based largely on cultural differences [25]. We elected to use this definition of generation because it contains information about culturally relevant changes with regards to mental illness, sex, and reproduction and simultaneously captures differences in participants' ages. Our approach is similar to prior studies of natural selection which have also used categorical definitions of birth cohorts [26,27]. Models using age or birth year instead of generation are provided in the (S1 and S2 Tables respectively) and do not conflict with conclusions presented here.

## Descriptive approaches

Means and standard deviations were calculated for number of male sexual partners, number of pregnancies, number of live births, PHQ-9 score, age, and body mass index (BMI) for the entire sample and for each generation. PHQ-9 score was calculated by summing responses to questions 1–9 from the scale and is a previously validated measure of depression severity [28]. BMI was calculated from respondents' reported height and weight. Continuous variables were compared among generations using separate ANOVAs in SAS version 9.4. We produced descriptive graphical representations of each fitness component as a function of birth year by applying locally estimated scatterplot smoothing (LOESS) in SigmaPlot 14.5.

Frequency information for demographic data was also calculated, including self-reported race/ethnicity, socioeconomic status, and level of education. Self-reported race/ethnicity is divided into five categories in NHANES: Mexican American, Non-Hispanic Black, Non-Hispanic White, Other Hispanic, and Other Race, including Multiracial. Level of education is divided into five categories: less than 9th grade education, 9-11th grade education or 12th grade education without a high-school diploma, high school graduate or equivalent, some college or associate (AA) degree, and college graduate or above. Due to differences in questionnaire items on socioeconomic status which prevented data harmonization across all survey years included in this study, socioeconomic status is included as a binary variable (family income above vs. below $20,000 per year, which is near the U.S. federal poverty cutoff for a 3-person family). Gender identity was not reliably collected and not presented here. Categorical variables were compared among generations using separate chi-square tests in SAS version 9.4.

## Relative fitness components

Relative measures of each fitness component were obtained by dividing an individual's fitness value by the mean value for that component for all participants included in the study. Lifetime number of male sexual partners, pregnancies, and live births were self-reported by participants in NHANES. We created measures of relative number of sexual partners, relative pregnancy success, and relative birth success, obtained following removal of outliers identified by examination of z-scores of the lifetime numbers provided (removed individuals with z-score>6). These different components of fitness reflect different episodes of selection [29].

We then standardized PHQ-9 score and relative fitness components to a mean of 0 and standard deviation of 1. We obtained standardized regression coefficients from the regression of each relative fitness measure on PHQ-9 score for each generation separately [29]. This approach allows us to compare the strength of the relationship between depressive symptoms and relative fitness components between generations. Non-linear selection was explored, but

because most patterns were not significant, we did not include these results (see Supporting information).

## Generalized linear models

To assess how other factors might influence the link between depressive symptoms and fitness components, we also evaluated this link using separate generalized linear models in proc genmod (SAS version 9.4). Visual inspection of the distribution of relative number of sexual partners, relative pregnancy success, and relative birth success revealed that all three variables were highly left-skewed with a large number of individuals with a relative fitness of zero for each of these components. We therefore fit generalized linear models with a Tweedie distribution and log link function [30].

For each generalized linear model, we included generation, BMI, race/ethnicity, level of education, and socioeconomic status as covariates. The highest two levels of education were found to be collinear with socioeconomic status. However, both variables were retained in the models because socioeconomic status is an important influence on psychiatric symptoms and its removal did not notably change statistics in the models (S3 Table) [31]. Given that age, racial/ethnic groups, income levels and education varied among generations, we analyzed the relationship between relative fitness components and PHQ-9 score by generation (Table 1). We also examined first-order interaction terms and retained significant interactions that lowered AIC for the final model because including all interactions or all significant overactions overfit our models. These models allow us to evaluate the relationship between depressive symptoms and fitness components even after controlling for variables that potentially confound the relationships described by the linear regression of relative fitness components on PHQ-9 score. The ratio of Pearson $\chi^2$ to degrees of freedom for all models indicated good model fit (for relative mating success, Pearson $\chi^2$:d.f = 2.08; for relative mating success, Pearson $\chi^2$:d.f = 1.05; for relative birth success, Pearson $\chi^2$:d.f = 0.81).

## Results

Contrary to widely held assumptions, there was consistently positive selection on depressive symptoms for all three fitness components; however, the strength of selection changed over time (Fig 1). The positive relationship between relative number of sexual partners and PHQ-9 score became progressively stronger across generations (Fig 1A): it was positive, but not significant, for the Silent Generation (Adjusted $R^2$ (Adj. $R^2$) = 0.0041, F = 3.50, standardized β ($\beta_{ST}$) = 0.076, df = 1, 602, $P$ = 0.062) and increasingly strong and positive for the Baby Boomer Generation (Adj. $R^2$ = 0.0096, F = 75.06, $\beta_{ST}$ = 0.10, df = 1, 3970, $P < 0.0001$), Generation X (Adj. $R^2$ = 0.022, F = 75.06, $\beta_{ST}$ = 0.15, df = 1, 3221, $P < 0.0001$), and the Millenial Generation (Adj. $R^2$ = 0.051, F = 120.05, $\beta_{ST}$ = 0.23, df = 1, 2229, $P < 0.0001$; Fig 1).

The relationship between relative pregnancy success and PHQ-9 score was consistently positive across all generations but varied in strength (Fig 1B), being strongest in the Baby Boomer Generation and slightly weaker in other generations (Silent Generation: Adj. $R^2$ = 0.0046, F = 3.81, $\beta_{ST}$ = 0.079, df = 1, 602, $P$ = 0.051, Baby Boomer Generation: Adj. $R^2$ = 0.019, F = 28.63, $\beta_{ST}$ = 0.14, df = 1, 3970, $P < 0.0001$, Generation X: Adj. $R^2$ = 0.017, F = 55.54, $\beta_{ST}$ = 0.13, df = 1, 3221, $P < 0.0001$, and the Millenial Generation: Adj. $R^2$ = 0.012, F = 28.63, $\beta_{ST}$ = 0.11, df = 1, 2229, $P < 0.0001$; Fig 1).

There was no relationship between relative birth success and PHQ-9 score (Fig 1C) for the Silent Generation (Adj. $R^2$ = -0.0015, F = 0.12, $\beta_{ST}$ = 0.014, df = 1, 602, $P$ = 0.73), but a weakly positive relationship for the Baby Boomer Generation (Adj. $R^2$ = 0.0093, F = 38.29, $\beta_{ST}$ = 0.098, df = 1, 3970, $P < 0.0001$), Generation X (Adj. $R^2$ = 0.0043, F = 14.78, $\beta_{ST}$ = 0.068, df = 1,

**Table 1. Descriptive statistics of participants.**

| Variable | | Overall (N = 10030) | Generation | | | | df | F or χ2 | p-value |
|---|---|---|---|---|---|---|---|---|---|
| | | | Silent Generation (N = 604) | Baby Boomer Generation (N = 3972) | Generation X (N = 3223) | Millenial Generation (N = 2231) | | | |
| PHQ-9 | | 3.8 (0.05) | 3.3 (0.2) | 4.1 (0.08) | 3.8 (0.08) | 3.4 (0.09) | 3 | 10.56 | <0.0001 |
| # Male Sexual Partners | | 6.8 (0.1) | 4.0 (0.3) | 6.5 (0.2) | 7.7 (0.2) | 7.0 (0.2) | 3 | 24.89 | <0.0001 |
| # Pregnancies | | 2.8 (0.02) | 3.4 (0.09) | 3.2 (0.03) | 3.0 (0.03) | 1.5 (0.04) | 3 | 402.15 | <0.0001 |
| # Live Births | | 2.0 (0.02) | 2.9 (0.08) | 2.4 (0.02) | 2.1 (0.02) | 1.1 (0.03) | 3 | 486.91 | <0.0001 |
| Age | | 43.6 (0.1) | 66.0 (0.09) | 54.9 (0.1) | 38.0 (0.1) | 25.5 (0.08) | 3 | 17958.50 | <0.0001 |
| BMI | | 28.6 (0.07) | 28.9 (0.2) | 29.5 (0.1) | 28.8 (0.1) | 27.1 (0.1) | 3 | 56.90 | <0.0001 |
| Race/ Ethnicity | Mexican American | 1645 (16.4) | 82 (13.6) | 601 (15.1) | 562 (17.4) | 400 (17.9) | 12 | 87.50 | <0.0001 |
| | Non-Hispanic Black | 2240 (22.3) | 139 (23.0) | 959 (24.1) | 679 (21.2) | 463 (20.7) | | | |
| | Non-Hispanic White | 4149 (41.4) | 282 (46.7) | 1652 (41.6) | 1346 (41.8) | 869 (38.9) | | | |
| | Other Hispanic | 1056 (10.5) | 73 (12.1) | 454 (11.4) | 301 (9.3) | 228 (10.2) | | | |
| | Other Race, including Multiracial | 940 (9.3) | 28 (4.64) | 306 (7.7) | 335 (10.4) | 271 (12.1) | | | |
| Family Income | <$20,000 | 2365 (23.6) | 183 (30.3) | 897 (22.6) | 619 (19.2) | 666 (29.8) | 3 | 100.25 | <0.0001 |
| | >$20,000 | 7665 (76.4) | 421 (69.7) | 3075 (77.4) | 26.04 (80.8) | 1565 (70.1) | | | |
| Level of Education | <9th Grade | 663 (6.6) | 83 (13.7) | 332 (8.4) | 181 (5.6) | 67 (3.0) | 12 | 199.39 | <0.0001 |
| | 9th-11th Grade or 12th Grade without Diploma | 1386 (13.8) | 100 (16.6) | 538 (13.5) | 449 (13.9) | 299 (13.4) | | | |
| | High School Graduate/ GED or Equivalent | 2113 (21.1) | 126 (20.9) | 920 (23.2) | 608 (18.9) | 459 (20.6) | | | |
| | Some College or AA Degree | 3381 (33.7) | 176 (29.1) | 1244 (31.3) | 1068 (33.1) | 893 (40.0) | | | |
| | College Graduate or Above | 2487 (24.8) | 119 (19.7) | 938 (23.6) | 917 (28.4) | 513 (23.0) | | | |

Mean (standard deviation) for PHQ-9 score, fitness components, age, and BMI and the count (percentage) for race/ethnicity, family income, and level of education for the entire sample and each generation.

3221, $P$ <0.0001), and the Millenial Generation (Adj. $R^2$ = 0.0069, F = 16.59, $\beta_{ST}$ = 0.086, df = 1, 2229, $P$ <0.0001; Fig 1).

Among outcome variables, PHQ-9 score differed between generations as did the number of male sexual partners, pregnancies, and live births in independent ANOVAs (Table 1). Overall, the number of male sexual partners appears to be relatively stable over the sample period, but the number of pregnancies and live births is consistent until the 1970s with a decline in these fitness components for women from Generation X and the Millenial Generation (Fig 2). Among important covariates, age differed between generations, as did BMI, frequency of respondents from different racial/ethnic groups differed significantly between generations frequency of respondents from different reported family income levels, and frequency of respondents with different reported levels of education (Table 1). Including these covariates in the models (generation, BMI, race/ethnicity, level of education, and family income) did not change our finding of positive selection on depressive symptoms (Table 2).

## Discussion

Mental illness can have devastating, life-long impacts on an individual's ability to function in various domains of life. Because of the disruptive effects of mental illness on interpersonal relationships, it is widely assumed that it will have negative fitness consequences [32]. Attempts to

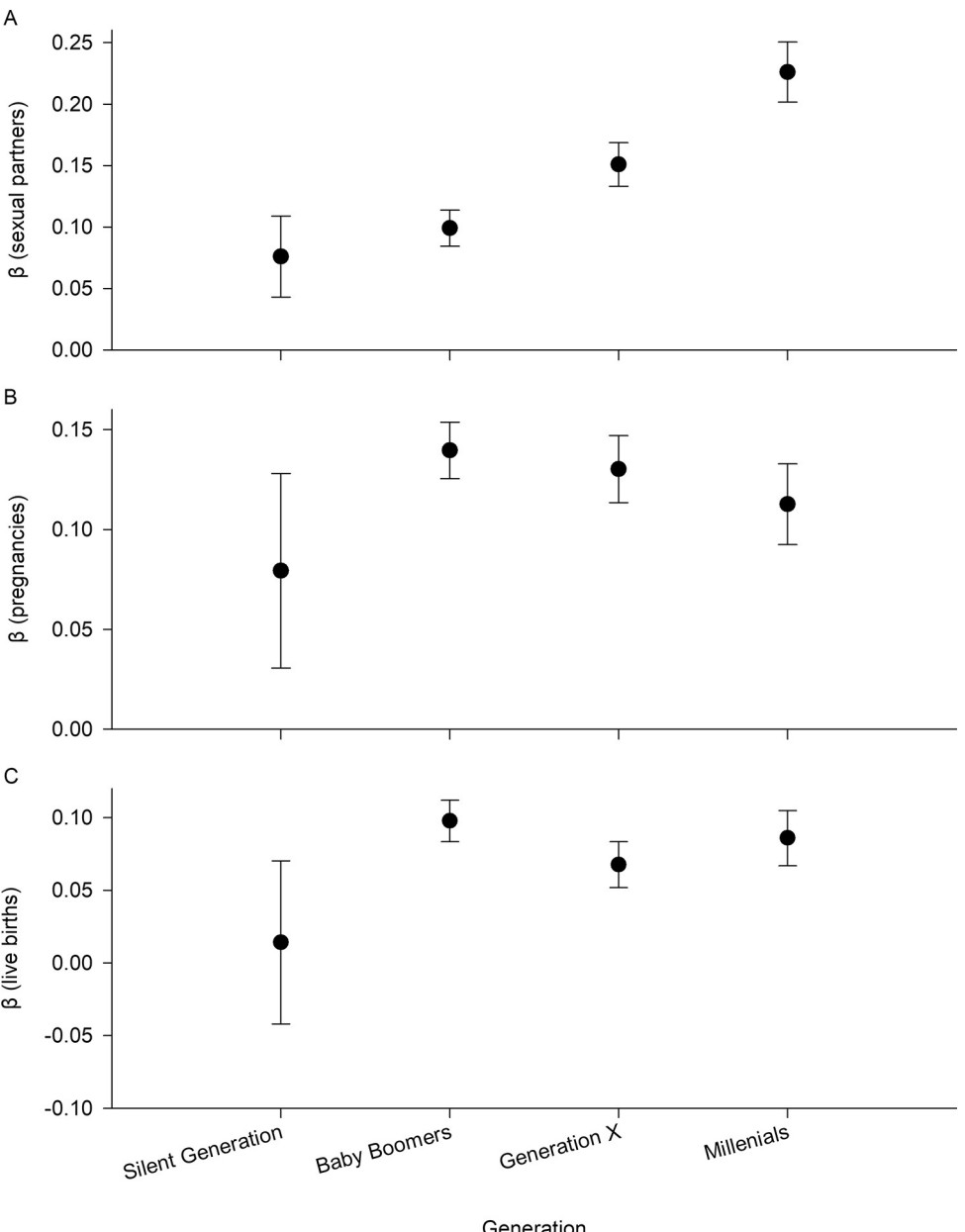

**Fig 1. Changes in selection on depressive symptoms (measured as PHQ-9 score) across generations for three components of fitness.** (A) relative number of male sexual partners, (B) relative pregnancy success, and (C) relative birth success. Shown are standardized regression coefficients (β) and their standard errors obtained from linear regression of each relative fitness component on PHQ-9 score for each generation separately. All regression coefficients were positive and significant except in the Silent Generation.

assess this assumption have found decreased fecundity in individuals with schizophrenia and bipolar disorder and early menopause in individuals with major depressive disorder [5,33]. Women with mental illness are also more likely to experience recurrent miscarriage [12]. Some studies have shown that individuals with depressive pathology have lower fecundity and are less likely to ever have children [9,11]. These patterns contrast with other findings that individuals with major depressive disorder have similar lifetime fecundity compared to unaffected siblings or the general population and may have an earlier age at first reproduction

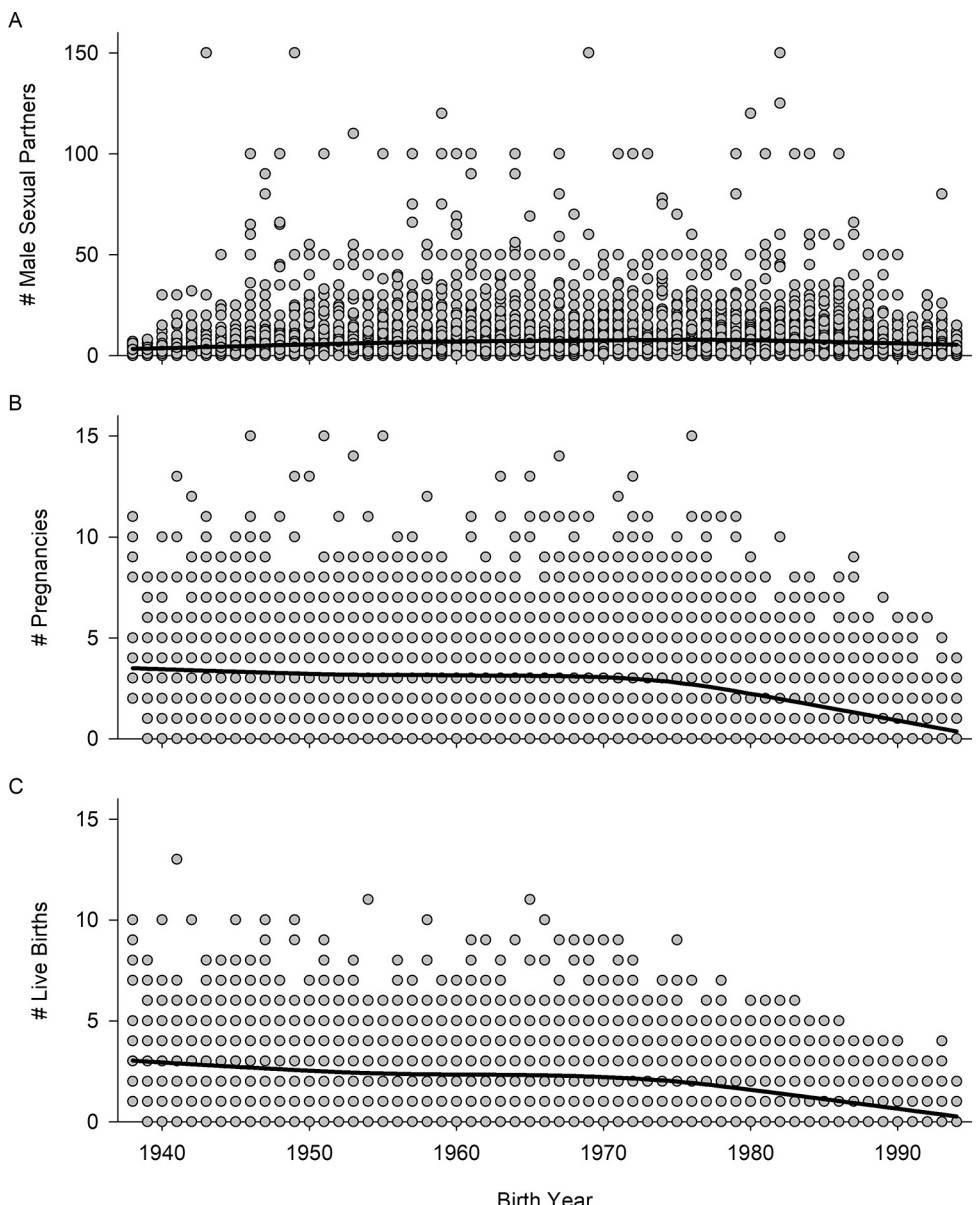

**Fig 2. Fitness components as a function of birth year.** Number of male sexual partners (A), number of pregnancies (B), and number of live births (C) plotted as a function of the respondents' birth years. Each plot shows a solid black line which was obtained by smoothing with the locally weighted scatterplot smoothing (LOESS) function in SigmaPlot 14.5 using a sampling proportion of 0.5 and a first-degree polynomial.

[5,8,11]. The symptoms of mental illness, however, overlap between diagnostic categories and are commonly experienced by individuals who do not meet criteria for any psychiatric diagnosis. Due to this phenotypic heterogeneity, prior work that focuses solely on the extreme expression of these traits can produce misleading results.

Here, using a unique long-term study that assessed depressive symptoms across a wide distribution of women, most of whom have never been diagnosed with mental illness, we found consistent evidence of positive selection on depressive symptoms using three separate metrics of fitness. These findings contrast with widely held assumptions and provide a new explanation for the maintenance of genetic variation in depression in human populations. Across the

**Table 2. Generalized linear models of depressive symptoms and fitness components.**

| Variable | Relative Mating Success (β = 0.042) | | | Relative Pregnancy Success (β = 0.011) | | | Relative Birth Success (β = 0.0036) | | |
|---|---|---|---|---|---|---|---|---|---|
| | d.f. | Chi-Square | p-value | d.f. | Chi-Square | p-value | d.f. | Chi-Square | p-value |
| PHQ-9 | 1 | 173.23 | <0.0001 | 1 | 58.2 | <0.0001 | 1 | 5.07 | <0.0001 |
| Generation | 3 | 20.38 | <0.0001 | 3 | 192.68 | <0.0001 | 3 | 167.19 | <0.0001 |
| BMI | 1 | 6.12 | 0.0133 | 1 | 27.55 | <0.0001 | 1 | 31.21 | <0.0001 |
| Race/Ethnicity | 4 | 109.02 | <0.0001 | 4 | 111.37 | <0.0001 | 4 | 106.44 | <0.0001 |
| Level of Education | 4 | 185.26 | <0.0001 | 4 | 508.85 | <0.0001 | 4 | 513.12 | <0.0001 |
| Family Income | 1 | 46.26 | <0.0001 | 1 | 6.23 | 0.0125 | 1 | 1.3 | 0.2538 |
| PHQ-9 x Generation | 3 | 24.64 | <0.0001 | | | | | | |
| Generation x BMI | 3 | 30.22 | <0.0001 | 3 | 39.37 | <0.0001 | 3 | 33.23 | <0.0001 |
| Generation x Level of Education | | | | 12 | 209.95 | <0.0001 | 12 | 131.36 | <0.0001 |
| BMI x Race/Ethnicity | 4 | 47.03 | <0.0001 | | | | | | |

Results of generalized linear models of the relationship between each fitness component and PHQ-9 corrected for important covariates and the interactions between these covariates. β indicates the strength of the relationship for each component.

20[th] Century, we show that women with more depressive symptoms tended to have more sexual partners, more pregnancies, and more live births (Fig 1). While this relationship was weaker in the Silent Generation, it was striking that we did not find negative selection on depressive symptoms for any fitness component in any generation. The relationship between depressive symptoms and fitness was also robust to inclusion of important covariates, including BMI, race/ethnicity, family income, and level of education as well as significant interaction terms (Table 2).

Our results suggest that fluctuations in the strength of selection may play a role in the maintenance of variation in depressive symptoms [34]. While we did not detect fluctuations in the direction of selection, the strength of this relationship did vary over time (Fig 1). For relative number of sexual partners, the relationship strengthened over time (Fig 1A). This finding is consistent with a recent Mendelian randomization study which demonstrated increased number of sexual partners causally worsened the risk of depression [35]. These observed patterns also raise the possibility of conflict between episodes of selection, where the relationship between depression and one fitness component may be strengthening while simultaneously not changing or weakening with another [29]. Most importantly, our results highlight the fact that negative fitness consequences of depressive symptoms cannot be simply assumed based on clinical distress or dysfunction. A possible explanation for why our results differ from prior results lies in the fact that we examined depressive symptoms across the population, as opposed to comparing individuals with and without a psychiatric syndrome. Our results should not be interpreted as indicative of positive selection on depressive syndromes, but rather as evidence of positive selection on depressive symptoms.

Alternatively, selection could be acting on some other unmeasured trait that is genetically associated with depression and mediating the link between depressive symptoms and fitness detected here (i.e. correlational selection) [36]. Since the symptoms of depression in PHQ-9 are genetically correlated with each other, this could be true for any individual component. Regulation by hormones, for example, can facilitate or constrain the response of traits to selection since hormonal pleiotropy can generate adaptive suites of genetically correlated traits [37]. Given the importance of hormonal regulation to various psychiatric disorders and person perception, this area of study may be particularly fruitful [38]. Likewise, telomere dynamics involved in reproduction, senescence, and mental health provide an alternative target of selection which may underlie the selective patterns detected in our results [39–41].

The retrospective, observational nature of our study limits inferences into the direction of causality between depressive symptoms and fitness consequences in women, but some mechanisms are worth considering here. First, our study spans multiple, important cultural changes that occurred during the 20[th] Century. Cultural influences on both depression and reproduction may be an important modulator of the relationship between the two. For example, pharmacological treatments for depression were not available until the 1950s and major improvements to this class of medication did not occur until the 1980s which could account for changes in strength of selection in later generations [42]. Another undoubtedly important change is a decline in fecundity in women born after 1972. This decline (Fig 2B and 2C) coincides with increased access to contraceptives and reproductive healthcare for women in the U. S. Mild fluctuations in the strength of selection may be partially influenced by a changing cultural milieu, which includes changing societal norms and attitudes toward mental illness, sex, and reproduction.

Second, depression may be mechanistically linked to hormonal changes in women associated with mood and reproduction. Risk of depression is known to be higher in women, a pattern which emerges in puberty and continues throughout life [43]. Other important life history events such as menopause are also related to risk for major depressive episodes [44]. Up to 16% of pregnant women will experience a major depressive episode [45]. Peripartum women are well-known to have increased risk of developing a depressive episode and risks mirror those observed in the general population [46,47]. Fertility disorders are also commonly associated with major depressive episodes [48]. Moreover, a strong link between mood disorders and reproductive events (e.g. menarche, pregnancy, parturition, and menopause) has been well-established [49]. Estrogen signaling plays a key role in regulation of the menstrual cycle and pregnancy while cessation of estrogen signaling is a key feature of menopause [50,51]. Estrogen simultaneously may regulate mood via its role in regulating the serotonergic and dopaminergic systems [52]. Fluctuations in estrogen during the menstrual cycle, after parturition, or with onset of menopause are thought to partly contribute to the known risk of mood disorders associated with these events [53]. Notably, systems downstream of the estrogen receptor are targeted by psychopharmacological interventions in affective disorders [54]. Stabilizing natural fluctuations in estrogen with exogenous administration of the sex hormone can also improve depressive symptoms in women [55]. Therefore, our results, in combination with the known links between hormonal aspects of female fertility cycles, may indicate that high fecundity and mood disorders are mechanistically linked where increased fecundity may play a causal role in producing depressive symptoms.

Another limitation of the present study is that part of our sample has not yet completed their reproductive years. The median age at which women have their last child is approximately 40-years-old and 90% of women have completed reproduction by age 45 [56]. In our sample, women in the Silent Generation and Baby Boomer Generation can reasonably be assumed to have completed reproduction. With a median age of 38-years-old in Generation X, most of these women will have nearly completed reproduction. The oldest woman born in the Millenial Generation in our sample is 35, suggesting significant future reproductive potential among women in this generation. This age-related effect is reflected in the lower average relative pregnancy success and relative birth success among these cohorts (Table 1). In the generalized linear models, however, the relationships between depressive symptoms and fitness components remained positive even after controlling for this generational difference (Table 2). Conclusions about the strength of the relationship between depressive symptoms and fitness components should be made with caution for these younger generations, but the finding of positive selection appears robust.

Our study is also limited by the measurement of depressive symptoms at a single timepoint. The severity of depressive symptoms measured during sampling may not reflect the average severity of symptoms at other times in an individual's life. However, evidence indicates that mood likely shows significant repeatability across the lifespan. First, approximately 50% of individuals with major depressive disorder have experienced their first major depressive episode before age 25 [57]. Given the median age of women in our sample, most individuals with major depressive disorder would have already experienced their first episode by the time they participated in NHANES interviews. Second, even among individuals being treated with antidepressants, the relapse rate (development of a major depressive episode after remission) nears 40% within 20 years [58].

## Conclusion

Overall, we have demonstrated that women with more depressive symptoms have a relatively higher number of sexual partners, pregnancies, and live births. The direction of this relationship was positive in each generation sampled, but the strength of the relationship fluctuated over time. This finding suggests that prior assumptions about the negative fitness consequences of depression are not valid, and that fluctuating selection may be an important mechanism for maintenance of variation in mental illness in human populations. Our findings also highlight the importance of measuring key psychological traits using the full spectrum of variation in the population, rather than focusing on extreme phenotypes that fall into discrete categories of disorders. We strongly urge future work on the evolution of mood disorders to consider both biological and cultural mechanisms that explicitly link depressive symptoms with fitness as well as changing societal influences on women's reproductive decisions.

## Supporting information

**S1 Table. Generalized linear models of depressive symptoms and fitness components by age.** Statistics from generalized linear models replacing generation (see Table 2) with age.
(DOC)

**S2 Table. Generalized linear models of depressive symptoms and fitness components by birth year.** Statistics from generalized linear models replacing generation (see Table 2) with birth year.
(DOC)

**S3 Table. Generalized linear models of depressive symptoms and fitness components with family income covariate removed.** Statistics from generalize linear models removing family income due to detected collinearity (compare with Table 2).
(DOC)

## Acknowledgments

The authors would like to thank Randolph M. Nesse for insightful discussions during conceptual development of this manuscript.

## Author Contributions

**Conceptualization:** Christopher I. Gurguis, Nicole M. Bucaro.

**Data curation:** Christopher I. Gurguis.

**Formal analysis:** Christopher I. Gurguis, Renée A. Duckworth.

**Investigation:** Christopher I. Gurguis.

**Methodology:** Christopher I. Gurguis, Renée A. Duckworth.

**Supervision:** Renée A. Duckworth, Consuelo Walss-Bass.

**Validation:** Renée A. Duckworth.

**Visualization:** Christopher I. Gurguis, Renée A. Duckworth.

**Writing – original draft:** Christopher I. Gurguis, Renée A. Duckworth, Nicole M. Bucaro, Consuelo Walss-Bass.

**Writing – review & editing:** Christopher I. Gurguis, Renée A. Duckworth, Nicole M. Bucaro, Consuelo Walss-Bass.

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
