## [Decision Letter · Decision Letter 0]

5 Jul 2024

PONE-D-24-07233Fitness consequences of depressive symptoms vary between generations: Evidence from a large cohort of women across the 20th centuryPLOS ONE

Dear Dr. Gurguis,

Thank you for submitting your manuscript to PLOS ONE. After careful consideration, we feel that it has merit but does not fully meet PLOS ONE’s publication criteria as it currently stands. Therefore, we invite you to submit a revised version of the manuscript that addresses the points raised during the review process.

We look forward to receiving your revised manuscript.

Kind regards,

Liliana G Ciobanu

Academic Editor

PLOS ONE

2. Please note that your Data Availability Statement is currently missing a direct link to access each database. If your manuscript is accepted for publication, you will be asked to provide these details on a very short timeline. We therefore suggest that you provide this information now, though we will not hold up the peer review process if you are unable.

Reviewers' comments:

Reviewer's Responses to Questions

**Comments to the Author**

1. Is the manuscript technically sound, and do the data support the conclusions?

Reviewer #1: Yes

2. Has the statistical analysis been performed appropriately and rigorously? 

Reviewer #1: Yes

3. Have the authors made all data underlying the findings in their manuscript fully available?

Reviewer #1: Yes

4. Is the manuscript presented in an intelligible fashion and written in standard English?

Reviewer #1: Yes

5. Review Comments to the Author

Reviewer #1: The authors use a large data set to test for associations between depressive symptoms and fitness components in women finding generally positive associations suggesting directional selection on depression. The manuscript is generally well-written, the analyses are appropriate and of general interest, but I have some concerns and comments regarding the limitations of the data, the model construction and the hypotheses being tested.

1) There is an inherent problem with the data being collected somewhat simultaneously, which means that differences between generations (which are to some extent defined arbitrarily) are confounded by differences in age and cumulative experiences. For example, lifetime number of offspring, partners etc. increase during life. It is probably premature to include the Millennial generation (which is still reproducing), and it makes the data interrupted by right censoring, which is not accounted for. It is probably also biased towards individuals with early onset of depressive symptoms, early reproduction etc. This is to a large extent acknowledged and discussed by the authors in the last part Discussion, but I suggest clarifying the major limitations early in the Abstract and Introduction.

2) More information on the PHQ-9 score and data is needed, i.e. I understand this covers depressive symptoms across two weeks. This raises the questions whether there is any individual reproducibility in this measure (can you test that?), if this is the result of only one (or more) questionnaires per individual, and to what extent PHQ-9 is influenced by other variables that may be confounding potential fitness components. Furthermore, is PHQ-9 known to be associated with age?

3) If it is correct that individuals with higher PHQ-9 scores have more severe mental disorders (e.g. schizophrenia, bipolar disorder etc., L75-85), it is relevant to test for non-linear (quadratic) relationships between PHQ-9 and fitness, which could further reveal if mild (and not strong) depressive symptoms are positively associated with fitness. It is not clear if the authors excluded individuals diagnosed with mental illness, and thus if depressive symptoms are confounded by other mental disorders.

4) The findings regarding fluctuations in the strength of selection over time are mainly based on the relationships being weaker or absent in the Silent Generation (e.g. L277), but the authors should consider that the sample size in that generation is much smaller than the others (and that earlier generations will have had more time to accumulate fitness components).

5) The authors should clarify that there may not necessarily be positive selection for depression symptoms based on the data, but that selection could instead be acting on some other (unmeasured) trait associated with depression and mediating links with fitness. Speculative ideas include telomere dynamics and hormone levels. Alternatively, a discussion of cause and effect could be raised (L297), i.e. are depressive symptoms more likely to occur before or after reproduction? Depressive symptoms may be caused by reproduction (L314), but in that case there is no positive selection, but a correlated response to selection on another trait.

6) The response to selection on a trait depends on the heritability of the trait and of the covariation between fitness and the trait. While the heritability of mental disorders may be considerable (L54-), it is not clear from the Introduction if depressive symptoms are heritable.

L41-43: Define generations by birth year here.

L54: Constant selection?

L115: Did you account for relatedness among participants?

L117: Did you exclude individuals that died while still at reproductive ages?

L179-194: How were models validated?

L184: Tweedie distribution is probably fine, but it may be relevant to test other zero-inflated distributions.

L186-189: Did you test for multicollinearity among predictor variables?

L189: I suggest you construct one global model where generation can be included as a covariate and interaction. You lose considerable statistical power by subsetting data by generation.

L212: All regression *coefficients.

Figure 2: What does the black line show? How have you (statistically) evaluated “relatively consistent” and “decreased”.

I suggest plotting the data and regressions showing fitness components vs. PHQ-9 for each generation in one figure.

The data used in this study (after exclusions) should be made available or identifiable in the repository.

I suggest including the code in the supporting information.

6. PLOS authors have the option to publish the peer review history of their article (what does this mean?). If published, this will include your full peer review and any attached files.

Reviewer #1: No

---

## [Author Response · Author response to Decision Letter 0]

18 Aug 2024

The authors would like to thank the Reviewer for helpful comments on our manuscript. We truly believe that the comments have helped to improve the quality of the manuscript. We have done our best to incorporate suggestions into the body of the text where possible. In some cases, we have also added to the supporting information. For each of the Reviewer’s points below, our replies are given in italics. Line numbers reference the file with Track Changes enabled. Thank you again for taking the time to review our paper.

Reviewer #1: The authors use a large data set to test for associations between depressive symptoms and fitness components in women finding generally positive associations suggesting directional selection on depression. The manuscript is generally well-written, the analyses are appropriate and of general interest, but I have some concerns and comments regarding the limitations of the data, the model construction and the hypotheses being tested.

1) There is an inherent problem with the data being collected somewhat simultaneously, which means that differences between generations (which are to some extent defined arbitrarily) are confounded by differences in age and cumulative experiences. For example, lifetime number of offspring, partners etc. increase during life. It is probably premature to include the Millennial generation (which is still reproducing), and it makes the data interrupted by right censoring, which is not accounted for. It is probably also biased towards individuals with early onset of depressive symptoms, early reproduction etc. This is to a large extent acknowledged and discussed by the authors in the last part Discussion, but I suggest clarifying the major limitations early in the Abstract and Introduction.

Author’s reply: We agree with the inherent problem identified by the Reviewer and, as noted by the Reviewer, have addressed this in detail in the Discussion. Given the importance of this limitation, we have added lines highlighting it in both the Abstract (L46-47) and Introduction (L102-105) as suggested.

2) More information on the PHQ-9 score and data is needed, i.e. I understand this covers depressive symptoms across two weeks. This raises the questions whether there is any individual reproducibility in this measure (can you test that?), if this is the result of only one (or more) questionnaires per individual, and to what extent PHQ-9 is influenced by other variables that may be confounding potential fitness components. Furthermore, is PHQ-9 known to be associated with age?

Author’s reply: A paragraph addressing repeatability/individual variation in PHQ-9, the single measurement of PHQ-9 in NHANES, and examinations of the relationship between PHQ-9, age, and other sociodemographic variables has been added to the Methods section (L115-124). In brief, a recent meta-analysis shows that PHQ-9 is repeatable over short periods of up to 2 weeks, but data examining repeatability over longer periods is unavailable. PHQ-9 in NHANES has been shown to have stable distribution with age and shows invariance for sex, race/ethnicity, and education level.

3) If it is correct that individuals with higher PHQ-9 scores have more severe mental disorders (e.g. schizophrenia, bipolar disorder etc., L75-85), it is relevant to test for non-linear (quadratic) relationships between PHQ-9 and fitness, which could further reveal if mild (and not strong) depressive symptoms are positively associated with fitness. It is not clear if the authors excluded individuals diagnosed with mental illness, and thus if depressive symptoms are confounded by other mental disorders.

Author’s reply: Individuals with diagnosis of psychiatric disorders were not excluded from the analyses and the Reviewer’s point is well-taken (now clarified in L140). In earlier versions of the manuscript, to examine for the presence of non-linear selection, quadratic relationships between PHQ-9 and fitness were included in models. This term was significant for the number of sexual partners and live births in a single generation, but not for the number of pregnancies in any generation. Since one of the major goals of this manuscript is to scrutinize the hypothesis that symptoms of mental illness are always deleterious, the results of these models were not included in the final draft of the manuscript. We have now included these results in the supporting information and directed interested readers there (L196-197).

4) The findings regarding fluctuations in the strength of selection over time are mainly based on the relationships being weaker or absent in the Silent Generation (e.g. L277), but the authors should consider that the sample size in that generation is much smaller than the others (and that earlier generations will have had more time to accumulate fitness components).

Author’s reply: While we do note that the sample size for the Silent Generation is much smaller than others, at N = 604, it is alone much larger than other studies measuring phenotypic selection (see Kingsolver et al. 2001’s review of the strength of phenotypic selection in the wild in which most studies’ sample size was <135). Thus, given our overall large sample sizes for every generation, we have substantial power for detecting differences in standardized selection gradients between generations. Further, our results are robust to alternative analyses which use age or birth year in place of generational cohorts (see supporting information). Finally, while the Reviewer is correct that the Silent Generation have had more time to accumulate fitness components, The Baby Boomer Generation’s average age in our sample is 54.9 (older than the median age at menopause) and Generation X’s average age in our sample is 38.0. The median age of menopause is around 51 years old and the median age at last birth for women is around 31. Given this, we think that our fitness estimates for these generations are accurate. The only generation that may be missing some accumulated fitness is the Millennial Generation (average age 25.5 in our study) and omitting them this generation from the analysis does not change the fact that selection changes from the Silent to Baby Boomer generations. We have also acknowledged and addressed in detail the limitation arising from the cross-sectional nature of our study relevant to these points in the Discussion (L371-394).

5) The authors should clarify that there may not necessarily be positive selection for depression symptoms based on the data, but that selection could instead be acting on some other (unmeasured) trait associated with depression and mediating links with fitness. Speculative ideas include telomere dynamics and hormone levels. Alternatively, a discussion of cause and effect could be raised (L297), i.e. are depressive symptoms more likely to occur before or after reproduction? Depressive symptoms may be caused by reproduction (L314), but in that case there is no positive selection, but a correlated response to selection on another trait.

Author’s reply: We heartily agree with this point! A paragraph has been added to the discussion to highlight the possible relevance of correlational selection for interpretation of our results (L327-336). We also agree that the causal direction is important for understanding the pattern detected here and emphasize this in the discussion (L337-349).

6) The response to selection on a trait depends on the heritability of the trait and of the covariation between fitness and the trait. While the heritability of mental disorders may be considerable (L54-), it is not clear from the Introduction if depressive symptoms are heritable.

Author’s reply: We agree with this conceptual point and have added a line commenting on the heritability of depressive symptoms (L77-79). We have additionally added a line about the heritability of PHQ-9 in particular to the Methods section in the same paragraph in which we added relevant details about the PHQ-9 (see above and L122-124)).

L41-43: Define generations by birth year here.

Author’s reply: Years defining the four generational cohorts were added (L32-33)

L54: Constant selection?

Author’s reply: we have added the word consistent (L55)

L115: Did you account for relatedness among participants?

Author’s reply: A citation demonstrating the presence of cryptic relatedness in NHANES data collected between 1988-1994 and 1999-2002 has been added to the text. Since we were unable to account for relatedness in our data specifically, we have added a line acknowledging that it is a possible limitation in our dataset in the Methods (L143-147).

L117: Did you exclude individuals that died while still at reproductive ages?

Author’s reply: We did not exclude individuals that died while still at reproductive age as this information is not available in the database. Moreover, lifetime reproductive success as a measure of fitness is influenced by both fecundity and survival, so even if we could exclude individuals that died before menopause, we would not. 

L179-194: How were models validated?

Author’s reply: The ratio of Pearson χ2 to degrees of freedom for each generalized linear model was close to 1, indicating good model fit (a line has now been added to include these numbers, L219-221). Competing models were evaluated by comparing AICs (L214-216).

L184: Tweedie distribution is probably fine, but it may be relevant to test other zero-inflated distributions.

Author’s reply: We did consider the use of other zero-inflated distributions (Poisson and negative binomial), however, Tweedie was selected in the end because relative fitness is a ratio variable rather than a count and because women who reported no sexual partners or who have never been pregnant may be different in some important respect with regard to reproductive physiology.

L186-189: Did you test for multicollinearity among predictor variables?

Author’s reply: Collinearity among variables was assessed in our generalized linear models using the method described here for proc genmod in SAS (https://support.sas.com/kb/32/471.html). Collinearity among categorical variables and between categorical and continuous variables was assessed by first creating dummy variables and applying the procedure referenced. We have now detailed our approach in the supporting information and added a line to the manuscript to indicate where to find our procedure and code (L208-212). A new table has been added to the supporting information and its title and legend to the main manuscript file (L592-594). Our model results are robust to removal of family income which was found to be partly collinear with the two highest levels of education (see supporting information).

L189: I suggest you construct one global model where generation can be included as a covariate and interaction. You lose considerable statistical power by subsetting data by generation.

Author’s reply: Firstly, as discussed in our reply to point #4 above, our sample size is significantly larger than those available for other studies of phenotypic selection and subsetting the data by generation is important since 1. one of our aims is to look for the presence of fluctuating selections and 2. All quantitative genetic parameters (heritability, selection, etc.) are unique to each generation and should not be assumed to remain stable through time. For these reasons, standardized selection coefficients were obtained for each generation separately.

Secondly, a global model as described has already been included in the paper for each fitness component with generation included as a covariate and interaction term with other sociodemographic covariates (reported in Table 2) and alternative models using age and birth year in place of generation have been reported in the supporting information. The inclusion of this model allows us to examine the direction and magnitude of selection on PHQ-9 score while controlling for important covariates. Both types of models (examining generations independently or together at once in a single model) produce the same patterns of results highlighting the robustness of our results.

L212: All regression *coefficients.

Author’s reply: Thank you – correction made (now appears at L239)!

Figure 2: What does the black line show? How have you (statistically) evaluated “relatively consistent” and “decreased”.

Author’s reply: The black line is obtained from LOESS in SigmaPlot. The figure legend has been updated to reflect how we obtained the figure rather than our interpretation of it (L267-273).

I suggest plotting the data and regressions showing fitness components vs. PHQ-9 for each generation in one figure.

Author’s reply: Due to the complexity of the global model constructed to evaluate the relationship between fitness components, PHQ-9 score, and sociodemographic covariates, we elected to present the statistics which would be associated with the suggested plot as a table (Table 2). We strongly feel that the requested plot would be redundant on the information already presented in Table 2.

The data used in this study (after exclusions) should be made available or identifiable in the repository.

Author’s reply: We will make our data available.

I suggest including the code in the supporting information.

Author’s reply: The code for analyses has now been included in the supporting information.

A typo was found on lines 589 and 591 of the manuscript and in the legend for S1 and S2 Tables in the supporting information. The text for these tables mistakenly referenced Table 1 when it should have referenced Table 2. This has now been corrected.

---

## [Editor Report · Decision Letter 1]

3 Sep 2024

Fitness consequences of depressive symptoms vary between generations: Evidence from a large cohort of women across the 20th century

PONE-D-24-07233R1

Dear Dr. Gurguis,

We’re pleased to inform you that your manuscript has been judged scientifically suitable for publication and will be formally accepted for publication once it meets all outstanding technical requirements.

Kind regards,

Liliana G Ciobanu

Academic Editor

PLOS ONE
---

## [Editor Report · Acceptance letter]

21 Sep 2024

PONE-D-24-07233R1 

PLOS ONE

Dear Dr. Gurguis, 

I'm pleased to inform you that your manuscript has been deemed suitable for publication in PLOS ONE. Congratulations! Your manuscript is now being handed over to our production team.

Kind regards, 

on behalf of

Dr. Liliana G Ciobanu 

Academic Editor

PLOS ONE